# LeViS: A Vision Transformer for Fast Combinatorial Optimization of Imaging Techniques

## Abstract

The performance of a vision model depends greatly on the types of data on which it is trained. For applications from classifying objects to predicting the weather, models can be trained on a variety of image sensors and sampling patterns. Vision systems often face practical constraints (e.g. acquisition time, compute resources, energy, and memory consumption) that limit the number of sensors and samples. In such cases, obtaining and training the model on a subset of available modalities is beneficial. Although many iterative combinatorial optimization algorithms can find optimal sets of modalities, they are drastically slowed by the need to train the model to evaluate each proposed set. We introduce LeViS (Learned Vision System), a vision transformer that proficiently classifies images collected with any sampling pattern or sensor set *without retraining*. The predictive power of a set can of techniques can thus be quickly determined by evaluating the performance of LeViS on the set. We use LeViS with a variety of optimization algorithms (genetic algorithms, beam search, simulated annealing, sequential) to rapidly find optimal sets of satellite wavelength channels to classify local climate zones using So2Sat, and optimal pixel sampling patterns to classify handwritten digits and charaters using MNIST. Evaluating sets with LeViS instead of training new models enables optimization algorithms to find optimal sampling patterns and sensor sets up to 6800x faster.

## 1 Introduction

For many applications of computer vision, especially in scientific and industrial settings, many imaging techniques and sampling patterns are applicable. Microscopes, for example, can image hundreds of channels, using brightfield, phase contrast, fluorescent markers, and Raman spectroscopy to analyze the morphology and behavior of cells (Fischer et al., 2011; Pinkard et al., 2024; Bray et al., 2016; Zumbusch et al., 1999). Hyperspectral satellites image the Earth for years with hundreds of wavelength bands for agricultural, military, environmental, and civil applications (Zhu et al., 2020; Bhargava et al., 2024). Robotic perception systems commonly use RGB and IR cameras, ultrasound, lidar, radar, and haptics to map the environment (Liu et al., 2024).

Beyond choosing which imaging methods to use, vision systems must choose how to sample the world with each method. Microscopy methods such as confocal microscopy or two-photon microscopy, as well as macroscopic imaging methods such as lidar, steer the focus of a laser to build 3D images by scanning one point at a time. Robots must choose where to point their cameras. MRI machines measure points of the Fourier transform of the 3D image of a subject using radio waves (Zbontar et al., 2019).

Although we would like to use every imaging method to measure every point in space, we are often constrained in the amount of data we can image. For example, many microscopy methods require harsh chemicals or light, so live cells can only be imaged with a few imaging channels (Icha et al., 2017). When monitoring very fast microscopic dynamics such as neurons firing in brain tissue, scientists only have time to image only a few points in space since point-scanning the whole volume is too slow (Li et al., 2024). Hyperspectral satellites like NASA's Hyperspectral Infrared Imager satellite can measure 5.2TB/day, necessitating careful management of transmission bandwidth, power, and storage (Sun & Du, 2019). Mobile vision systems such as drones, robots, or self-driving cars have limited space, power, and computation, so the placement and number of cameras is essential.

Thus, to determine the constrained set of imaging techniques or sampling patterns that maximizes the performance of a task (e.g., classifciation, segmentation) when a model is trained on it, it is common to first collect and analyze an *anchor dataset* which contains subjects imaged with a wide variety of fully-sampled imaging methods. The optimal subset determined using the anchor dataset can then be deployed in the constrained environment. For example, fully-sampled MRI datasets can be used to determine fast scanning patterns (Zbontar et al., 2019). Biologists can image a single plate of cells with dozens of microscopy methods to determine the few methods to deploy in a drug discovery pipeline.

However, while exhaustive search is not practical due to rapid combinatorial explosion (~126 trillion ways to choose 25 methods out of 50), there are a plethora of highly effective combinatorial optimization algorithms, such as genetic algorithms, simulated annealing, sequential selection, and beam search, which iteratively evaluate and propose sets to arrive at an optimum.

Unfortunately, evaluating each set of imaging methods is expensive, as vision models are usually trained to use a single set of channels; a CNN trained on RGB images will not work with x-ray images. For each proposed set of imaging techniques, the model needs to be retrained, making exploration of the design space computationally prohibitive. Many iterative combinatorial algorithms are thus impractical as they would require training hundreds of models.

In this work, we leverage the dimensional flexibility of the attention mechanism (Vaswani et al., 2023) to create a model that optimally classifies images regardless of the imaging techniques or spatial sampling pattern used to collect the image. To accomplish this, LeViS constructs patch tokens for each imaging method separately, encodes the tokens from each method into an compressed latent representation, then cross-attends to the latents to produce a prediciton. If LeViS is trained on a single patch pattern or set of channels, it is unable to generalize to different combinations of channels and sampling patterns at test time. To solve this, we randomly mask patches in the image or whole channels during training. This way LeViS to learns to opportunistically attend to all relevant information in the input and generate accurate predictions for arbitrary combinations of imaging techniques.

By encoding each imaging method separately and using cross-attention against the latents to generate a prediction, LeViS's computational requirements scale linearly with the size of the sampling pattern and the number of imaging methods used. LeViS can also cache the latent representations for each imaging method, allowing inference with different combinations of imaging methods on the anchor dataset by only running the decoder. Inference using cached latents in practice takes less than half the time of running the whole model, drastically accelerating combinatorial optimization.

## 2 RELATED WORK

**Vision transformers (ViT)** have proved to be one of the most powerful architectures for various computer vision tasks (Dosovitskiy et al., 2020; Carion et al., 2020). Researchers have applied ViT for a variety of applications, including inferring gene function from multichannel microscopy, classifying deforestation using satellite imagery, and segmenting 3D MRI images, to name a few (Sivanandan et al., 2023; Kaselimi et al., 2023; Hatamizadeh et al., 2022).

A persistent issue with transformer architectures remains the quadratic scaling of the self-attention mechanism (Vaswani et al., 2023). This is especially problematic as the number of pixels scales quadratically with the height or width of 2D images and linearly with the number of imaging methods or channels, meaning that imaging applications with multi-modal imaging applications of transformers have massive computational requirements.

The Perceiver and Perceiver IO architectures use an asymmetric cross-attention mechanism to project large inputs into fixed-sized latents, then operate on the latents, to linear scaling with the size of the input (Jaegle et al., 2021; 2022). Perceivers support multimodal inputs, though modalities cannot be dynamically added or removed. Masked autoencoders completely decouple computational requirements from image size during training by selecting a random subset of patches from the image to operate on, but use the full image during inference (He et al., 2021).

**Masked image encoding methods** learn to process images corrupted by masking. Denoising autoencoders (DAE) have use a convolutional neural network (CNN) to denoise images masked by

noise (Vincent et al.). Context encoders and masked autoencoders leverage CNNs and transformers respectively to reconstruct arbitrary regions or patches removed from an image (Pathak et al., 2016; He et al., 2021). For hyperspectral imaging data, masked transformers have also been used to reconstruct random spatial-spectral blocks removed from satellite hyperspectral images (Scheibenreif et al., 2023). We note that spatial sampling of an image can easily be posed as masking all unsampled points in the image.

Models can also leverage masking in the channel dimension. Most machine learning models are purpose-built for a set of channels and do not function with only a subset of channels or with different channels. Strategies to make models channel-adaptive include training the model to expect all possible channels and inputting synthetic random data for unseen channels or tokenizing each channel of the input separately and adding a learned channel embedding to each token or learning convolutional weights and regularization parameters that can be applied to any channels (Shetab Boushehri et al., 2024; Bao et al., 2024; Chen et al., 2023).

**Methods to choose sets of imaging techniques** can be generally divided into three categories: analytical methods, searching methods, and embedded methods. Analytical methods analyze the statistics each imaging method directly to determine which to select. These include simple ranking methods that look at statistics of single pixel values such as variance, contrast, signal-to-noise ratio (SNR), and entropy to choose imaging methods or spectral bands (Liu et al., 2018; Chang et al., 1999), analyzing correlations between pixels of different imaging methods (MartÍnez-UsÓMartinez-Uso et al., 2007). Analytical methods do not take into consideration how combinations of channels perform with the model that analyzes them.

Searching methods iteratively propose a combination and test how well a model works with the combination. Many classical optimization algorithms are applicable to guide the search, including probabilistic search algorithms such as genetic algorithms and simulated annealing, heuristic search algorithms like beam search and hill climbing, and sequential search algorithms. Other more recent combinatorial optimization methods, such as using a neural solver Gao et al. (2025) or using combinatorial bayesian optimization Oh et al. (2019); Deshwal et al. (2023) are also applicable and potentially more sample efficient. The largest downside of searching methods is the expense of each evaluation, requiring retraining a model. Some methods of imaging technique optimization attempt to avoid retraining by observing the average effect of many random corruptions of each channel, but these techniques only give information about single-channel statistics, and require more than 100 evaluations of the whole dataset for each imaging technique to get reliable statistics.

Embedded methods have emerged as a promising technique to select imaging methods. In one strategy, candidate imaging channels are partitioned and ranked to isolate the most informative subset. These reduced bands are then passed through a hemispherical reflectance-based spatial filter and a 3D CNN, achieving highly effective classification (Phaneendra Kumar et al., 2024). Another method proposes a joint deep learning architecture composed of a constrained measurement learning network followed by a classifier, where the network directly learns to select bands that maximize task performance rather than relying on hand-crafted metrics (Ayna et al., 2023). Methods such as GFSNetwork employ temperature-controlled Gumbel-Sigmoid sampling to automatically determine informative subsets during training, offering a scalable solution that improves efficiency while preserving interpretability (Wydmański & Śmieja, 2025). Researchers have also analyzed the attention patterns of attention-based convolutional networks trained on all possible channels, then selected the subset of channels that were most attended to Ribalta Lorenzo et al. (2020).

## 3 METHOD

Given an input imaged with any collection of imaging methods or an arbitrary pixel sampling patterns, LeViS is designed to generate the best possible output for the data that it is given. This allows us to use LeViS as a fast evaluation function for the effectiveness of different sets of imaging methods or spatial sampling patterns without training a new model from scratch, greatly accelerating a wide variety of combinatorial optimization algorithms.

In practice, using any subset of channels or pixel-patterns in an image, LeViS can either (i) classify the input or (ii) reconstruct the original image. This dual function supports both **hypothesis-driven** optimization (e.g., optimizing classification performance for a specific task such as classification

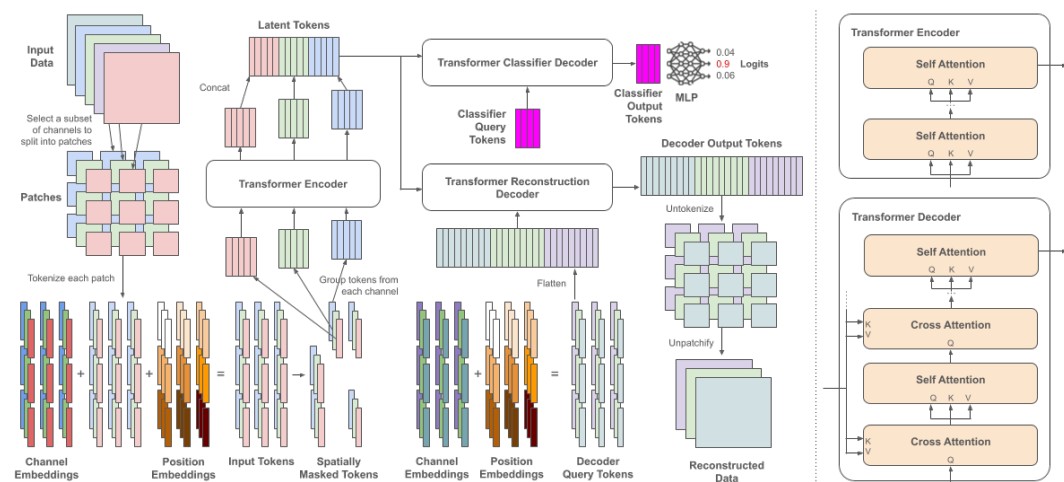

Figure 1: LeViS architecture overview

or segmentation) and **hypothesis-free** optimization (e.g., identifying sampling patterns maximally representative for the raw data).

## 3.1 LeViS Model Architecture

**Channel Masking:** Consider an input image $x$ with dimensions $H \times W \times C$ from the anchor dataset, with spatial dimensions $H \times W$ and imaged using $C$ imaging methods or channels. During training, a random number $C_{in} \in [1, C]$ is selected uniformly at random and $C_{in}$ channels from the image are randomly selected yielding a $H \times W \times C_{in}$ image to be input into the model. This dropout procedure, introduced by Bao et al. (2024), regularizes the model to produce stable results no matter which set of imaging methods are inputted into the model.

**Tokenizer:** Each channel of the image is separately split into non-overlapping square patches of $P \times P$ pixels, yielding $C_{in}$ sequences of $H//P \cdot W//P$ patches. A small convolutional neural network is applied to each patch to produce a token. As is standard in vision transformers, a sine-cosine position embedding is added to each token based on its position of origin(Dosovitskiy et al., 2020). A learned channel embedding is added to each token based on its channel of origin as in Bao et al. (2024) (Figure 1).

**Spatial Masking:** During training, a random number $n_{in} \in [1, H//P \cdot W//P]$ is selected uniformly at random, and all tokens in same $n_{in}$ positions in each of the $C_{in}$ sequences are deleted. This amounts to a randomly applied spatial mask on the image, similar to the masking procedure introduced in He et al. (2021).

**Encoder:** Tokens from each channel in the input are fed through the self-attention layers of the encoder separately to create a latent representation of each channel, which are concatenated and sent to the decoder. Thus, each token in the latent space arises from *a single channel*. Self-attention is applied to each channel separately, meaning that encoder computation scales linearly with the number of imaging methods used. Any combination of channels may be inputted, including single channels, so the latent representation of each channel must learn to contain all potentially relevant information for the decoder even if the information is redundant to other channels.

**Classifier Decoder:** A classification token and register tokens are passed through a series of alternating cross-attention layers (attending to the latent tokens) and self-attention layers. The classification token is fed into an MLP that produces class logits.

**Reconstruction Decoder:** Similarly to the classifier decoder, a set of learned query tokens, one for each patch in each channel, is fed through a series of cross-attention then self-attention layers. Each query token is then fed through a convolutional neural network decoder to produce a square patch,

and all patches are assembled into an output image. The reconstruction decoder can be used for self-supervised pre-training of LeViS or hypothesis-free discovery of sets.

## 3.2 SET OPTIMIZATION

After training, LeViS can be used rapidly evaluate the predictive power of different combinations of imaging methods or pixel sampling patterns by applying the combination or pattern to the validation section of the anchor dataset and scoring the performance. Since LeViS is trained to extract whatever semantically relevant is in the input regardless of the channels or spatial patches it contains, the validation performance is an excellent scoring function for the performance of the combination or pattern. LeViS can be used as a rapid evaluator to accelerate almost any iterative combinatorial optimization algorithm. In this paper we demonstrate optimization of the set of $k$ imaging methods or spatial patches of $n$ available on the following algorithms.

**Single-Shot** In single-shot optimization, we take the individually best-performing $k$ channels, and combine them into a set. This approach is highly efficient, but typically leads to lower accuracy. A key limitation is redundancy: features carrying similar information may be needlessly included. Conversely, features that perform poorly in isolation but complement each other when combined are likely to be overlooked.

**Forward Selection** This is a greedy, bottom-up feature selection procedure. It begins with an empty set of features and iteratively adds the feature that provides the best performance when combined with the previously selected features, continuing until $k$ features have been included. An issue with this and similar approaches is that they do not take into account complementary sets. The best set of size $k-1$ is not necessarily a subset of $k$.

**Backward Selection** A greedy, top-down procedure. It begins with a full set of features, and iteratively removes the feature that would least decrease performance, until $k$ features are left. This is similar to forward selection, with similar pitfalls - the ideal set for $k+1$ may not contain the ideal set for $k$.

**Genetic Algorithms** Begin with a random population of feature subsets, each of size $k$. At each generation, the performance of these subsets is evaluated, and the top-performing ones are retained. New subsets are produced by combining elements from two parent subsets (crossover) and occasionally altering features at random (mutation). Over successive generations, this process allows the population to evolve toward better solutions. The randomness of crossover and mutation helps the search escape local optima.

**Simulated Annealing** Start with an initial solution and a high "temperature," allowing the algorithm to accept both better and worse candidate solutions, encouraging exploration. As the temperature decreases, the probability of accepting worse solutions diminishes, making the search more greedy over time. This method moves past locally suboptimal feature sets to discover combinations that work well together.

**Beam Search** A more general forward selection, with parameter $b$, the beam width. At the first iteration, keep the best $b$ single-feature sets. At each subsequent step, all possible single-feature extensions of these $b$ sets are considered, and the best $b$ among them are retained. After $k$ steps, return the bestLarger beam widths allow the search to explore a broader set of candidates, increasing the chance of recovering near-optimal feature combinations that would be missed under a narrow beam, at the cost of higher computational complexity.

## 4 EXPERIMENTS

We evaluate LeViS on two image classification benchmarks: So2Sat (Zhu et al., 2020) and MNIST (Lecun et al., 1998), demonstrating its effectiveness for rapid combinatorial optimization of imaging methods and sampling patterns, respectively. So2Sat was chosen for its diversity of imaging methods, with 18 different satellite synthetic aperture radar and multispectral optical imaging methods used to capture each image, and its small size, which is representative of a typical anchor dataset. Labels of local climate zones (e.g., compact high rise buildings, dense trees, water) were assigned by hand by a team of experts. MNIST was chosen for its clear spatial structure, making spatial sam-

pling a meaningful and explainable task, and small size, representative of a typical anchor dataset. Each image contains a handwritten arabic numeral, 0 through 9.

## 4.1 OPTIMIZING COMBINATIONS OF IMAGING METHODS

For So2Sat we trained LeViS with 4x4 pixel patches, 2 self-attention layers in the encoder, and 1 cross-attention layer and 4 self-attention layers in the classifer. We used the random split of So2Sat, training for 3000 epochs over 48 hours on a server with 2 Nvidia RTX 5090 GPUs, achieving 98.8% Top-1 accuracy. For comparison, we achieved 97.81% accuracy with ResNet50 which has 25.6 million parameters, Zhu et al. (2020) achieved 97.82% accuracy using ViT-S/8 with 21 million parameters, and Bao et al. (2024) report 99.10% accuracy using ChannelViT-S/8 with 21 million parameters.

Trivially, the validation accuracy that LeViS attains given a set of imaging methods provides a lower bound on the utility of that set, as LeViS can be used for that set out of the box. However, if we train a LeViS on a single set of imaging methods without masking, the validation accuracy has the potential to marginally improve over the base LeViS model since part of the base model's representational capacity is used to support channels not in the set.

To test whether the LeViS's average validation accuracy is representative of the true utility of the set, we first generate 100 random combinations of 5 and 10 methods. We train the base LeViS model on all possible channels with the standard random masking procedure. Then for each set of methods, we compare the validation performance of the base model with the base model fine-tuned for 10 epochs, the rough number of epochs of fine tuning needed for training accuracy to converge (Figure 2).

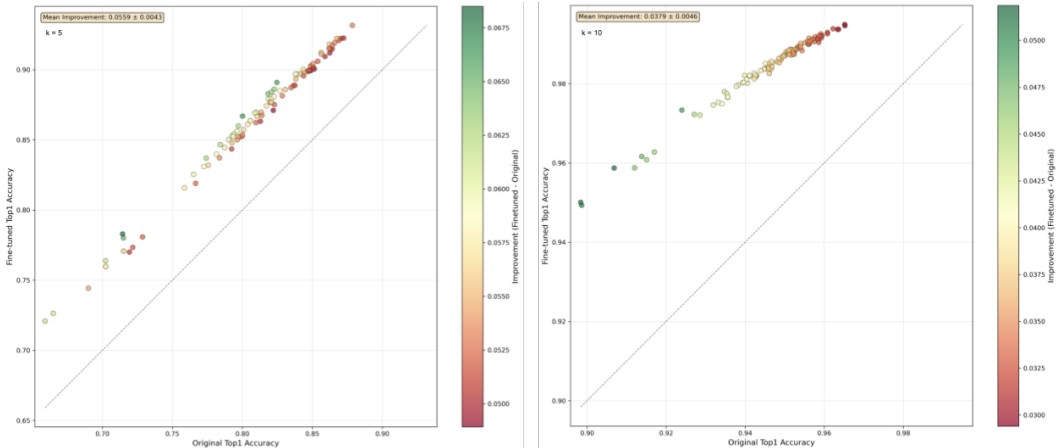

Figure 2: LeViS base accuracy vs fine-tuned accuracy on 100 random sets of 5 imaging methods (left) and 10 imaging methods (right)

Though we see modest increases in validation accuracies across all the combinations (avg +5.59% k=5, +3.79% k=10), the ordinality of scores produced by LeViS produces is largely equivalent to the scores of the fine-tuned model. This means that the validation scores generated by LeViS are almost always an accurate proxy to compare the predictive power of sets of imaging methods.

Leveraging LeViS as an ultrafast evaluator for different sets of imaging methods, we can quickly execute a variety of different iterative optimization algorithms. We first test a variety of optimization algorithms to find the optimal set of 5 channels from 18, with 8568 possible combinations we can quite easily compare the results to the optimum determined through exhaustive search (Table 1). We see that single-shot selection is extremely fast, as it simply picks the best-performing channels individually, but accuracy suffers significantly. Other methods find near-optimal solutions but take a few minutes. Notably, genetic algorithms and beam search perform the best, though less than a percent worse than the other methods. Most hyperparameters of the genetic algorithm are intuitive;

Table 1: Comparison of algorithms for selecting 5 channels of 18 from So2Sat

| Algorithm | Channels | Best Top-1 Accuracy | Runtime (s) |
|---|---|---|---|
| Single Shot | [9, 11, 10, 17, 16] | 0.7627 | 34.9 |
| Simulated Annealing T=2 100 iters | [2, 4, 9, 13, 17] | 0.8828 | 2087.4 |
| Simulated Annealing T=1 100 iters | [2, 4, 9, 13, 17] | 0.8828 | 2095.6 |
| Simulated Annealing T=2 200 iters | [4, 5, 10, 12, 17] | 0.8738 | 419.3 |
| Simulated Annealing T=1 200 iters | [1, 5, 9, 13, 16] | 0.8822 | 416.3 |
| Genetic Algorithm pop=16 500 gens | [3, 4, 8, 12, 17] | 0.8799 | 253.1 |
| Genetic Algorithm pop=16 1000 gens | [3, 4, 8, 12, 17] | 0.8799 | 160.3 |
| Genetic Algorithm pop=4 500 gens | [3, 8, 10, 15, 16] | 0.8722 | 91.8 |
| Genetic Algorithm pop=4 20 gens | [3, 8, 10, 14, 16] | 0.8716 | 51.0 |
| Beam Search b=3 | [1, 5, 9, 13, 17] | 0.8860 | 410.7 |
| Forward Selection | [9, 14, 5, 17, 1] | 0.8827 | 159.3 |
| Backward Selection | [3, 4, 8, 15, 17] | 0.8789 | 370.2 |
| Exhaustive search | [1, 5, 9, 13, 17] | 0.8860 | 8938.5 |
| Baseline | All channels | 98.77 | 2.14 |

more generations and larger populations yield better results. Beam search, though the most effective, takes much longer than all other methods with little marginal benefit.

The optimization time trials were run on a desktop computer with a Nvidia RTX 4090 GPU and AMD CPU with 32 cores. To run inference on the entire So2Sat validation dataset without using cached latents, LeViS took 5.52 seconds and around 13 GB of VRAM with a batch size of 512. For the same task using cached latents, LeViS took 1.96 seconds and 1.09 GB of VRAM, saving considerable time and computational resources. For comparison, training the ResNet50 architecture He et al. (2015) which has a similar number of parameters to LeViS (25.6 million vs 17.9 million parameters, respectively) to 97.8% accuracy on the same computer took 2 hours and 28 minutes. This means that training a CNN model to evaluate a combination of imaging methods is more than 5000x slower than using LeViS.

## 4.2 Optimization of Sampling Patterns for Accuracy

For MNIST, we trained LeViS with 2x2 pixel patches, 2 self-attention layers in the encoder, and 2 cross-attention and 4-attention layers in the decoder, totaling 735 thousand parameters. We trained LeViS for 1000 epochs over 56 minutes on a desktop computer with a Nvidia RTX 4090 GPU, achieving a 98.78% accuracy with all patches.

Similar to our imaging method experiment, to test whether the LeViS's average validation accuracy is representative of the true utility of the set of patches inputted, we first generate 100 random combinations of 30 2x2 pixel patches out of the 28x28 image. We train the base LeViS model using the random masking procedure. Then for each set of patches, we compare the validation performance of the base model with the base model fine-tuned for 10 epochs (Figure 3).

Though we see a greater increase in validation accuracies across all the combinations (avg +9.25% k=30), the ordinality of scores produced by LeViS produces is remains largely equivalent to the scores of the fine-tuned model. This means that the validation scores generated by LeViS are almost always an accurate proxy to compare the predictive power of spatial sampling patterns.

We evaluated a variety of combinatorial optimization methods to optimize sampling patterns for MNIST digit classification using LeViS. We constrained all methods to use just 28 2x2 patches from the 196 patches available. This results in a much larger combinatorial space than with So2Sat, with upwards of trillions of possible combinations (Table 2). For optimizing sampling patterns with MNIST, there is a more clear tradeoff between runtime Top-1 Accuracy. Genetic algorithms punch above their weight, attaining a Top-1 accuracy within half of a percent of the best Top-1 accuracy with a small fraction of the runtime. Notably, LeViS only takes 490 milliseconds to process the entire MNIST validation dataset, saving 56 minutes of training time for each evaluation.

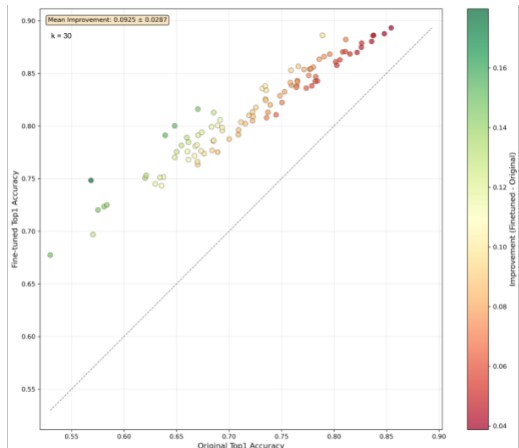

Figure 3: LeViS base accuracy vs fine-tuned accuracy on 100 random sets of 30 patches, MNIST

Table 2: Comparison of algorithms for selecting 30 2x2 pixel patches of 196 patches of MNIST

| Algorithm | Best Top-1 Accuracy | Runtime (s) |
|---|---|---|
| Single Shot | 0.8484 | 85.8 |
| Simulated Annealing t=0.02 1000 iterations | 0.9439 | 443.5 |
| Simulated Annealing t=0.02 200 iterations | 0.9176 | 89.9 |
| Simulated Annealing t=0.01 1000 iterations | 0.9494 | 446.6 |
| Simulated Annealing t=0.01 200 iterations | 0.9134 | 88.7 |
| Simulated Annealing t=0.005 1000 iterations | 0.9467 | 450.1 |
| Simulated Annealing t=0.005 200 iterations | 0.9391 | 88.3 |
| Genetic Algorithm pop=16 500 gens mutation=0.05 | 0.9459 | 399.6 |
| Genetic Algorithm pop=16 1000 gens mutation=0.05 | 0.9527 | 567.1 |
| Genetic Algorithm pop=8 500 gens mutation=0.05 | 0.9432 | 273.4 |
| Genetic Algorithm pop=8 1000 gens mutation=0.05 | 0.9475 | 327.9 |
| Genetic Algorithm pop=4 500 gens mutation=0.05 | 0.9244 | 86.4 |
| Genetic Algorithm pop=4 1000 gens mutation=0.05 | 0.9359 | 128.5 |
| Beam Search width=3 | 0.9555 | 6968.4 |
| Forward Selection | 0.9541 | 2520.3 |
| Backward Selection | 0.9558 | 10866.5 |
| Baseline | 0.9878 | 0.48 |

### 4.3 HYPOTHESIS-FREE OPTIMIZATION OF SAMPLING PATTERNS

Our goal with hypothesis-free optimization of sampling patterns is the search for the subset of pixel patches that contains the most information about the image as a whole. In practice, we search for the patch subset that can best reconstruct the whole image. We train LeViS on MNIST with 2x2 pixel patches, 2 self-attention layers in the encoder, and 2 cross-attention and 4-attention layers in the reconstruction decoder, totaling 1.1 million parameters. Training for 1000 epochs over 1.5 hours, we attain a final validation reconstruction MSE of 0.43. Using a genetic algorithm with population size 16 over 200 generations to optimize 30 patches, we obtain a reconstruction MSE of 0.715, which translates to a Top-1 accuracy of 79.2%.

## 5 CONCLUSION

In conclusion, our proposed method LeViS, vastly accelerates the combinatorial optimization of imaging methods and spatial sampling patterns for computer vision by eliminating the need to re-train a model to evaluate each proposed set. We demonstrate state of the art performance on two

image classification benchmarks: So2Sat (Zhu et al., 2020) and MNIST (Lecun et al., 1998), while reducing evaluation costs by several orders of magnitude. By serving as a fast, flexible evaluator, LeViS enables classical search strategies, such as genetic algorithms, beam search, and simulated annealing, to operate effectively in domains where exhaustive retraining is computationally infeasible. Beyond the tasks considered here, our framework opens the door to applying combinatorial optimization in broader scientific and industrial settings, including microscopy, robotics, and medical imaging. Future directions include extending LeViS to larger-scale multimodal datasets, integrating it with active data acquisition systems, and exploring its potential for real-time adaptive sensing. Overall, LeViS provides a practical foundation for rethinking how imaging pipelines are designed under real-world constraints.

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

# A APPENDIX

## A.1 STATEMENT ON USE OF AI TOOLS

Anthropic Claude was used to aid and polish writing in this paper.

