# OpenReview forum: "LeViS: A Vision Transformer for Fast Combinatorial Optimization of Imaging Techniques"
_ICLR.cc/2026/Conference — ICLR 2026 Conference Withdrawn Submission_

### Official Review · Reviewer_aX7M · 2025-10-27

**Soundness:** 2
**Presentation:** 1
**Contribution:** 3
**Rating:** 4
**Confidence:** 3

**Summary:**

This paper introduces LeViS, a vision transformer designed to classify images collected with any arbitrary combination of imaging methods (channels) or spatial sampling patterns without requiring retraining. The core problem is the computational expense of evaluating different subsets of imaging techniques for constrained vision systems, as traditional methods require training a new model for each proposed subset. LeViS addresses this by using a novel architecture that separately encodes tokens from each input channel, applies random channel and spatial masking during training, and uses a cross-attention-based decoder. This enables the model to generalize to unseen combinations of channels and patches at test time. The authors demonstrate that LeViS can be used as a fast evaluator with various combinatorial optimization algorithms (e.g., genetic algorithms, beam search) to find optimal sets, achieving speedups of over 5000x compared to retraining a CNN on the So2Sat and MNIST benchmarks.

**Strengths:**

-The core idea of creating a single, flexible model to evaluate arbitrary subsets of imaging modalities or sampling patterns without retraining addresses a significant bottleneck in practical system design. This is a novel and impactful contribution. The dual-use capability for both classification (hypothesis-driven) and reconstruction (hypothesis-free) optimization broadens the method's applicability . In addition, the concept of caching latent representations to drastically accelerate inference for combinatorial optimization is an interesting engineering insight.

-The proposed method LeVis is technically sound. The use of random channel and spatial masking during training is a well-motivated and effective strategy to force the model to become robust to any input subset.

**Weaknesses:**

-The paper describes the model architecture procedurally but lacks a clear, consolidated mathematical presentation. Key equations for the forward pass, the attention mechanisms, and the loss functions used for training (both classification and reconstruction) are missing (see Sec. 3.1). At the same time, the paper lacks descriptions of training details and the parameters used. This makes it difficult for subsequent work to follow, because based on the paper’s statements readers can hardly understand how to implement the method, for example what the model outputs are, where and what kinds of losses should be added, and how the outputs are used downstream to solve the problem that the paper focuses on.

-The experimental setup is limited., where some significant baselines are missing. While related work on channel-adaptive models is discussed (see Sec. 2), a direct quantitative comparison of LeViS's optimization performance against these specific baselines (e.g., [Bao et al., 2024]) is not provided. The performance of LeViS as a classifier is compared, but not its core contribution as an optimizer. The limitations of the "proxy evaluator" approach are acknowledged but not deeply analyzed. For instance, the conditions under which the LeViS score might fail to correlate with fine-tuned performance are not explored (see Sec. 4.1, 4.2).

-Additionally, the experiments in this paper make it difficult to quickly understand its practical value. The datasets used are too limited, and there is a lack of visualizations to explain what the proposed method can achieve. In other words, I cannot quickly grasp from the experiments what intuitive improvements LeVis brings to existing computer vision tasks.

-Finally, in addition to deficiencies in textual presentation, the preparation of figures and tables also has major problems. For example, in Table 1 the accuracy column in the last row shows an obvious numerical error, and Figures 2 and 3 are blurry with very small text, making them almost unreadable.

**Questions:**

-Add necessary formulas to explain the method’s data flow and training procedure to improve reproducibility.

-Add the referenced comparison baselines to verify the effectiveness of the proposed representation.

-I believe the experimental section requires major revision, as the current form makes it hard to understand practicality and concrete value; for example, add visualizations on datasets and concrete examples of applying the method to downstream tasks.

-Improve figure and table preparation and the writing; further proofreading and revisions are needed.

---

### Official Review · Reviewer_3Erx · 2025-10-30

**Soundness:** 2
**Presentation:** 2
**Contribution:** 3
**Rating:** 2
**Confidence:** 3

**Summary:**

The paper proposes LeViS, a Vision Transformer (ViT) architecture designed to address the bottleneck of retraining models in combinatorial optimization for imaging tasks. It enables adaptation to arbitrary imaging channels and sampling patterns without retraining, leveraging channel masking, spatial masking, and cross-attention mechanisms, along with a cached latent representation strategy for efficiency. Experiments span So2Sat (channel selection) and MNIST (sampling mode optimization), comparing against six optimization algorithms to demonstrate accuracy and speed improvements, with applications in microscopy, satellite imaging, and robotic perception.

**Strengths:**

The paper attempts to address a genuine pain point in imaging combinatorial optimization (i.e., costly retraining for each configuration). The technical design of LeViS, including channel/spatial masking and cross-attention, demonstrates a degree of technical sophistication. The experimental scope is broad, covering multiple datasets and optimization algorithms.

**Weaknesses:**

1.The graphs and charts are crude and unprofessional, and the experimental results do not support all the claims made by the authors.
2.There is a lack of richness in the comparative methods and a lack of systematic evaluation of existing methods.
3.The text is ambiguous and the writing structure is confusing.

**Questions:**

Please refer to weaknesses

---

### Official Review · Reviewer_UtWy · 2025-10-31

**Soundness:** 1
**Presentation:** 1
**Contribution:** 1
**Rating:** 2
**Confidence:** 4

**Summary:**

The key idea of LeViS, a vision transformer, is to train a transformer with random masking over channels and spatial patches, enabling it to classify or reconstruct images captured under arbitrary combinations of imaging methods without retraining.
Experiments show LeViS achieves near baseline accuracy while speeding up search by several orders of magnitude.

**Strengths:**

1. Addresses a practical bottleneck: the high cost of retraining during combinatorial search of imaging modalities.
2. Experimental results demonstrate that LeViS can indeed serve as a fast proxy evaluator for different combinations.

**Weaknesses:**

1. Lack of novelty or substantial contribution. The proposed model is a straightforward combination of standard ViT and existing random masking or channel dropout strategies (e.g., MAE, ChannelViT). No new architectural component, objective, or theoretical insight is provided.
2. Weak scientific motivation. The connection between combinatorial optimization and masked transformers is superficial.
3. No insight or analysis. The paper reports speedups but lacks deeper evaluation—no visualization, ablation on masking ratios, or analysis of model generalization to unseen sensor types.

**Questions:**

1. What is the essential novelty of LeViS compared to existing masked transformers such as MAE, Mask2Former, or ChannelViT?
2.  Could you explicitly clarify which component (architecture, objective, or training mechanism) is newly proposed rather than reusing existing ideas?
3. Could the authors mathematically explain how this framework optimizes or searches over combinations, rather than merely evaluating them?
4. Is the model able to generalize to unseen sensor configurations or only to subsets of known modalities?
5. The comparison baselines (genetic algorithm, beam search, etc.) are traditional search heuristics rather than recent learning-based optimization frameworks. Could you justify why these are considered sufficient to demonstrate LeViS’s advantage?

---

### Official Review · Reviewer_eK4G · 2025-11-01

**Soundness:** 2
**Presentation:** 2
**Contribution:** 2
**Rating:** 2
**Confidence:** 4

**Summary:**

This paper introduces a vision transformer-based framework for combinatorial optimization of imaging techniques. The core idea is to train a single model that can process arbitrary subsets of channels or spatial patches, and then use this pre-trained model as a fast proxy evaluator within classical optimization algorithms. The authors claim that this approach can speed up the search for optimal subsets by several orders of magnitude compared to the naive baseline of retraining a model for each candidate

**Strengths:**

The most impressive strength of this work is the large speedup it brings to the combinatorial optimization process, after training on an anchor dataset

**Weaknesses:**

- Its main motivation seems to be based on an argument that the only alternative is to retrain a model for every possible subset. This view ignores that many existing research in deep learning solves subset optimization using end-to-end differentiable approaches. For instance, in remote sensing, many studies use networks with learnable modules to directly find the best set of hyperspectral bands within a single training run. Similarly, in computational photography, other work has shown how to use networks to adaptively choose sensing parameters, like camera spectral sensitivities[1] , as part of the model itself. These methods completely avoid the repetitive retraining loop that the authors claim is the only other option. The proposed "train-then-search" method may actually be less efficient than these existing end-to-end solutions.
- The proposed LeViS framework, seem more like a good application of vision transformer, does not seem to solve a fundamental problem or offer deep insights. The contribution feels more like a trick to accelerate a specific workflow. In addition, the paper lacks a analysis of why this approach works. For example, there are no theoretical guarantees or bounds on the correlation between the proxy scores from LeViS and the true performance of a retrained model. The core assumption of ordinal equivalence is supported only by empirical evidence.
- The workflow has a significant cost, since the initial training of the universal model on the full anchor dataset is likely to be much more complex and computationally expensive than training a standard task-specific model, as it needs to learn a much more generalized representation space. The paper does not provide a clear comparison of this total training cost versus the cost of running an end-to-end differentiable subset selection method

[1]Joint camera spectral sensitivity selection and hyperspectral image recovery, ECCV 2018

**Questions:**

The core assumption is that the performance ranking provided by the universal model is a faithful proxy for the ranking of models specifically fine-tuned on each subset. While the empirical correlation is shown, what is the theoretical justification for this?

For other questions, please refer to my comments in Weakness part.

---

### Note · Authors · 2025-12-03

**Comment:**

We withdraw the paper as it needs additional work to clearly demonstrate it's efficacy. We thank the reviewers for their helpful comments and suggestions.

**Withdrawal Confirmation:**

I have read and agree with the venue's withdrawal policy on behalf of myself and my co-authors.